# SimVLG: Simple and Efficient Pretraining of Visual Language Generative Models

## Abstract

In this paper, we propose "SimVLG", a streamlined framework for the pre-training of computationally intensive vision-language generative models, leveraging frozen pre-trained large language models (LLMs). The prevailing paradigm in vision-language pre-training (VLP) typically involves a two-stage optimization process: an initial resource-intensive phase dedicated to general-purpose vision-language representation learning, aimed at extracting and consolidating pertinent visual features, followed by a subsequent phase focusing on end-to-end alignment between visual and linguistic modalities. Our one-stage, single-loss framework circumvents the aforementioned computationally demanding first stage of training by gradually merging similar visual tokens during training. This gradual merging process effectively compacts the visual information while preserving the richness of semantic content, leading to fast convergence without sacrificing performance. Our experiments show that our approach can speed up the training of vision-language models by a factor $\times 5$ without noticeable impact on the overall performance. Additionally, we show that our models can achieve comparable performance to current vision-language models with only $1/10$ of the data. Finally, we demonstrate how our image-text models can be easily adapted to video-language generative tasks through a novel soft attentive temporal token merging modules.

## 1 Introduction

The landscape of vision-language modeling has undergone significant transformations in recent years, with CLIP (Radford et al., 2021) serving as a landmark development. It distinguished itself through unparalleled zero-shot classification capabilities and efficiency in image-text retrieval tasks. Successive models like ALBEF (Li et al., 2021a), X-VLM (Zeng et al., 2022), and VLMo (Bao et al., 2022) further broadened the scope, addressing a myriad of tasks such as retrieval, visual entailment, and closed-set Visual Question Answering (VQA), among others.

Recently, the field has been enriched by the advent of generative models designed for complex image-to-language tasks. Notable contributions include CoCa (Yu et al., 2022), SimVLM (Wang et al., 2022d), Frozen (Tsimpoukelli et al., 2021), Flamingo Alayrac et al. (2022) and BLIP-2 (Li et al., 2023a), targeting tasks like image and video captioning and open-set VQA. Specifically, CoCa demonstrates robust performance across both uni-modal and multi-modal tasks, leveraging a large-scale dataset for training from scratch.

The computationally intensive nature of pre-training Vision-Language Models (VLMs) led to the conceptualization of BLIP-2. This model seeks to alleviate computational costs by employing pre-trained vision encoders (ViT) and language decoders (LLM). As illustrated in Figure 1a, a central innovation in BLIP-2 is the *Q-former*, a vision-language connector outfitted with learnable queries for effective cross-attention mechanisms. This architectural choice, however, demands an intensive pre-training regimen, referred to as *BLIP-2's Stage 1*. The stage involves three learning objectives—image-text contrastive, image-text matching, and language generation—and necessitates multiple forward passes for optimization.

Despite its efficiency gains over CoCa, BLIP-2's training still imposes considerable computational costs. This poses challenges for research environments with limited computational resources, such as university labs. Our experiments indicate that the Stage-1 training of BLIP-2 took approximately eight days on eight A100-80G GPUs. This computational burden has consequently restricted research

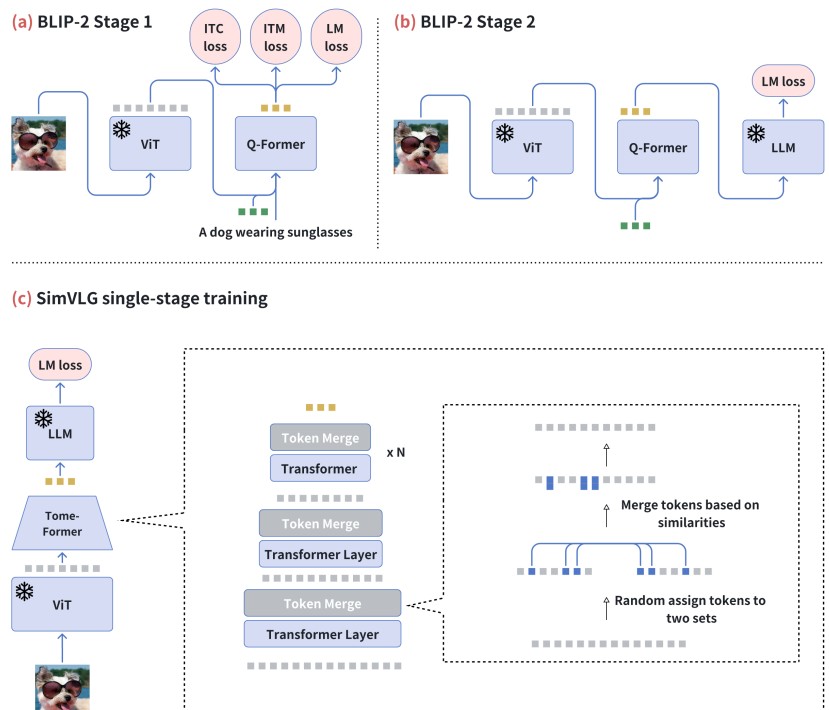

Figure 1: Overview of the BLIP-2 and SimVLG. **(a)** The 1st stage of BLIP-2 training involves a complex and computationally intensive process. It employs learnable queries (depicted in green) to select 32 tokens (in orange) from an extensive pool of 256 visual features (shown in grey). This queried output is then utilized to fulfill three distinct learning objectives, necessitating multiple forward passes within a single optimization step. **(b)** The 2nd stage of BLIP-2 adopts a conventional end-to-end training approach, mapping images directly to text. **(c)** In contrast, SimVLG employs a streamlined, single-stage training mechanism with a unified loss. Here, visual tokens (in grey) are progressively aggregated based on their inherent similarities at each layer of the TomeFormer architecture. The final set of merged tokens (in orange) serves as semantically rich but computationally efficient soft prompts, guiding the LLM to generate a corresponding caption for the input image.

to using the pre-trained Q-former, hindering the exploration of alternative ViTs in VLMs. This limitation is evident in subsequent works like InstructBLIP (Dai et al., 2023), VideoChat (Li et al., 2023b), Video-LLaMA (Zhang et al., 2023), X-LLM (Chen et al., 2023).

The prospect of reducing BLIP-2's computational cost through end-to-end, single-stage training is compelling. Such an approach would remove the complexities associated with resource allocation and hyper-parameter tuning inherent in multi-stage training. Yet, direct end-to-end training with BLIP-2 poses substantial challenges, corroborated by both original findings from BLIP-2 and our own empirical analyses. We hypothesize that these challenges emanate from the intrinsic design of the Q-former. Specifically, the inclusion of randomly initialized learnable queries and cross-attention mechanisms complicates the optimization landscape, especially when the aim is to minimize the representational disparity between visual and linguistic modalities.

In this paper, we propose an alternative to the Q-former, employing a systematic token-merging (Bolya et al., 2023) strategy that is both intuitive and effective. Here, token merging signifies the step-wise aggregation of tokens with analogous features across the layers of the Transformer model (see Figure 1c). We substitute the Q-former in BLIP-2 with a standard Transformer architecture augmented with token merging capabilities, which we term *TomeFormer*. Importantly, the TomeFormer is trainable to function effectively as an efficient vision-language connector. This modification, which we call **Sim**ple **V**isual **L**anguage **G**enerative pre-training model (SimVLG), facilitates a streamlined, single-stage training process. It requires only a singular learning objective and a single forward pass per optimization step. This stands in contrast to BLIP-2's multi-stage training, laden with multiple objectives and several forward passes.

Further, we introduce a *soft attentive temporal* token fusion mechanism within the ViT for effective video-language modeling. This eliminates the need for modality realignment, contrasting approaches such as the temporal Q-former (Zhang et al., 2023), or the addition of new learnable temporal queries (Li et al., 2023b). Our strategy simplifies the optimization challenges tied to working with relatively smaller video-text datasets, compared to their image-text counterparts. Remarkably, we demonstrate that even without video pre-training, our temporal token fusion approach can effectively train robust video-language models. This differs from recent work in video-language models that depend on pre-training models using vast million-scale video-text datasets.

Our contributions are summarized as follows:

- We adapt token merging, initially designed to enhance ViT inference speed without training, to serve as a means for condensing semantically-rich visual features within the vision language connector. Concurrently, we present a novel temporal token merging scheme for video modeling.
- Our proposed image-text model featuring TomeFormer competes effectively with BLIP-2, while requiring just a fraction of the computational resources. Given the reliance on BLIP-2's pre-trained model in contemporary studies, our approach widens the exploratory scope for various ViTs.
- We introduce a straightforward spatial attentive temporal modeling technique that allows for the seamless adaptation of pre-trained image-text models to video-text tasks. This approach eliminates the need for complex modality re-alignment, a common requirement in alternative methods.

## 2 RELATED WORK

**Image-Language Models**  Vision-language models generally fall into two categories: dual-encoder models and fusion-encoder models. Pioneering works like CLIP (Radford et al., 2021) and ALIGN (Jia et al., 2021) serve as exemplary dual-encoder models, demonstrating exceptional performance in zero-shot classification tasks. These architectures also excel in image-text retrieval, as their features can be pre-computed and stored, allowing for efficient similarity score computation via dot-product operations. Fusion-encoder models (Lu et al., 2019; Tan & Bansal, 2019; Alayrac et al., 2022; Dou et al., 2022b; Li et al., 2022b; Dou et al., 2022a; Xu et al., 2022), such as ALBEF (Li et al., 2021a), mPLUG (Li et al., 2022a), X-VLM Zeng et al. (2022), and VLMo (Bao et al., 2022), employ cross-attention mechanisms to enable deep interactions between visual and linguistic features. Other designs include concatenating features of each modality before feeding them into a Transformer (Chen et al., 2020; Li et al., 2020; Zhang et al., 2021; Gan et al., 2020; Li et al., 2021b; Cho et al., 2021; Huang et al., 2020; 2021; Shen et al., 2022; Kamath et al., 2021; Yang et al., 2022; Wang et al., 2022b; Kim et al., 2021; Xue et al., 2021; Wang et al., 2022a;c). These models excel in complex tasks like closed-set Visual Question Answering (VQA) and visual entailment. CoCa (Yu et al., 2022), trained on billions of image-text pairs, represents a state-of-the-art approach in generative tasks like open VQA and visual captioning. To mitigate the computational demands of pre-training, BLIP-2 (Li et al., 2023a) employs frozen pre-trained ViT and LLM components, focusing on training a specialized connector between visual and linguistic modalities called the Q-former. Due to the computationally intensive nature of training BLIP-2, subsequent models in visual instruction (Dai et al., 2023; Zhu et al., 2023; Li et al., 2023b) have predominantly utilized the pre-trained Q-former, which is aligned with the `eva-vit-g` model supplied by BLIP-2.

**Video-Language Models**  While many image-text models can be adapted for video-text tasks through simple feature pooling (e.g., VideoCoCa (Yan et al., 2022)), the field has seen specialized models that incorporate temporal dynamics. Building on the foundation of BLIP-2, Video-LLaMA (Zhang et al., 2023) enhances its architecture by introducing additional temporal Q-former layers between the spatial Q-former and the LLM components of the original BLIP-2 model. Inspired by BLIP-2, most recent works such as VideoChat (Li et al., 2023b), PandaGPT (Su et al., 2023), Valley (Luo et al., 2023), and Video-ChatGPT (Muhammad Maaz & Khan, 2023) leverage frozen LLMs in their video-language models.

**Token Merging**  Token Merging (ToMe) (Bolya et al., 2023) aims to improve the inference speed of pre-trained ViTs without requiring re-training. At each Transformer layer, tokens are divided into two sets and subsequently merged based on similarity, effectively reducing the token count and thereby accelerating inference. This method maintains classification and generation quality.

In our work, we repurpose ToMe to condense the visual features used as language prompts in the LLM. We integrate a standard Transformer with ToMe capabilities, resulting in a model we term TomeFormer. This model serves as an effective connector between visual and language domains, preserving semantic richness while reducing token count. Importantly, this integration of ToMe does not introduce any additional parameters. Inspired by spatial ToMe, we introduce a novel soft temporal ToMe variant within the vision encoder, thereby adding temporal modeling capabilities to our image-text models.

## 3 METHODS

We begin by presenting our image-text model and then describe the adaptations made to this pre-trained model for video-related tasks.

### 3.1 PRELIMINARY

Three key models serve as the foundation for generative tasks in the image-text domain: CoCa, Flamingo, and BLIP-2. The high computational cost of training CoCa from scratch and the proprietary nature of Flamingo led us to adopt the BLIP-2 framework. BLIP-2 utilizes frozen pre-trained ViT and LLM for vision-language tasks. Notably, BLIP-2 has gained significant traction in the field due to its open-source availability, thereby influencing subsequent research in various projects such as InstructBLIP, VideoChat, Video-LLaMA, X-LLM, Valley, and Video-ChatGPT.

**BLIP-2 Framework**  In BLIP-2, a ViT serves as the vision encoder, ingesting images and outputting a set of 257 visual tokens. A specialized component, the Q-former, is then added to the ViT. This Q-former utilizes 32 learnable queries to selectively extract and transform the 32 most informative tokens from the ViT's output pool of 257 tokens. These 32 tokens are subsequently used as soft language prompts to guide the LLM in generating text that describes the image content.

Although it is theoretically possible to train BLIP-2 end-to-end, empirical evidence suggests that such an approach often yields suboptimal outcomes. Consequently, BLIP-2 employs a preliminary Stage-1 pre-training phase for both the ViT and Q-former. During this phase, three learning objectives—Image-Text Contrastive (ITC) loss, Image-Text Matching (ITM) loss, and Language Modeling (LM) loss—are optimized simultaneously. This Stage-1 training involves 250,000 steps, each requiring multiple forward passes due to the multiple learning objectives.

Despite being more efficient than CoCa, our observations reveal that BLIP-2's Stage-1 training still demands approximately eight days on a server equipped with eight A100-80G GPUs. This computational cost poses a constraint for subsequent research that aims to explore various ViT configurations, as each new configuration requires a complete restart of the Stage-1 training process.

### 3.2 SIMVLG-IMAGE

We introduce SimVLG-Image (abbreviated as SimVLG, and shown in Figure 1c), an optimized vision-language generative pre-training model. Similar to BLIP-2, SimVLG utilizes a ViT for visual encoding and an LLM for linguistic decoding. The key innovation is the incorporation of a standard Transformer, augmented with spatial Token Merging, to act as the connector between the visual and linguistic modalities.

Formally, our framework includes a vision encoder $E_{\text{vision}}$, which ingests an input image $I$ and encodes it into a fixed set of visual tokens: $[v_1, v_2, ...v_L] = E_{\text{vision}}(I)$. Here, $L$ denotes the number of image patches. Subsequently, we employ a Transformer equipped with token-merging modules, termed as *TomeFormer* ($T_{\text{v} \to \text{t}}$) as the vision-to-language connector. This module effectively compresses the token count: $[v'_1, v'_2, ...v'_{L'}] = T_{\text{v} \to \text{t}}(f_{\text{proj}_1}([v_1, v_2, ...v_L]))$. In this equation, $L'$ is considerably smaller than the initial token count $L^1$. The LLM decoder then employs these compressed tokens as soft prompts for text generation: output $= D_{\text{LLM}}(f_{\text{proj}_2}([v'_1, v'_2, ...v'_{L'}]))$. Projection functions $f_{\text{proj}_1}$ and $f_{\text{proj}_2}$ are used to ensure dimension compatibility. Three main advantages of using TomeFormer are:

---

[1]We merge 19 tokens at each layer of the TomeFormer. Thus, 256 visual tokens are reduced to 28 tokens. Ablation on the number of merged tokens at each layer is studied in Section 5

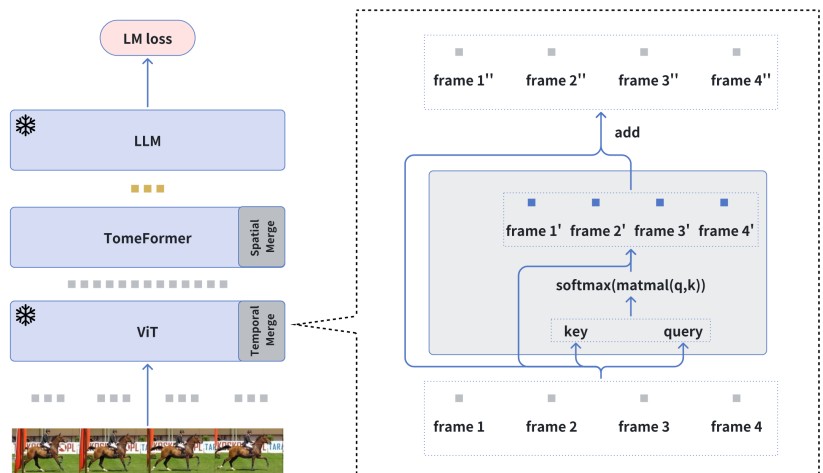

Figure 2: Overview of SimVLG-Video: In addition to TomeFormer's spatial token merging capabilities, our design introduces Temporal Attentive Soft Token Merging for nuanced temporal modeling. Each frame's output is calculated as a learnable weighted average of other frames in the video. This approach maintains the integrity of pre-existing, well-trained image-text models. For instance, when the input consists of static videos with identical frames, SimVLG-Video operates as if it were an image-text model. Importantly, this architecture avoids the need for complex model realignment, a requirement in alternative designs that insert a temporal Q-former between the visual encoder and the language model.

- Efficient token reduction, facilitating the transformation of loosely-structured visual data into a more concise yet informative representation.
- Computational efficiency, as the uncompressed ViT output consists of 256 tokens, plus a [CLS] token. Without compression, the subsequent vision-to-language connector would be computationally expensive in terms of both memory and processing power.
- Semantic richness of the compressed tokens. Unlike BLIP-2, which requires an extensive pre-training phase for feature extraction via its Q-former, TomeFormer naturally merges semantically similar tokens. Our empirical evidence confirms that TomeFormer-equipped models train more efficiently compared to alternatives like BLIP-2.

### 3.3 SIMVLG-VIDEO

Although many image-text models can be adapted for video-text tasks with minor modifications, they often overlook the importance of temporal modeling. For example, VideoCoCa extends CoCa using an attentive pooler without changing CoCa's architecture, while InstructBLIP and BLIP-2 employ a concatenated soft-prompt approach. This simplicity comes at the cost of inadequate temporal modeling, which is later addressed by VideoChat and Video-LLaMA through the introduction of learnable temporal queries and a temporal Q-former. However, these additions disrupt the integrity of the aligned VLM and necessitate a re-alignment process requiring substantial video-text pairs, as shown in VideoChat and Video-LLaMA.

In this paper, we propose a novel module called *Temporal Attentive Soft Token Merging* to enhance the ViT backbone with temporal modeling capabilities. Formally, let $v$ be a video feature tensor with dimensions $[B \times N \times L \times D]$, where $B$ is the batch size, $N$ is the number of frames, $L$ is the sequence length (i.e., the number of patches in a single video frame), and $D$ is the hidden dimension. Initially, we reshape $v$ into $[(B \times N) \times L \times D]$ which is subsequently fed into the self-attention layer of the ViT for *spatial modeling* as: $v' = \text{self-attn}(v.\text{reshape}(B \times N, L, D))$. For *temporal modeling*, $v'$ is reshaped to $[N, (B \times L), D]$. We then project this into key and query matrices $k$ and $q$ and compute $v''$ using our *Temporal Attentive Soft Token Merging* as follows: $k = W_{\text{key}}(v'.\text{reshape}(N, B \times L, D))$, $q = W_{\text{query}}(v'.\text{reshape}(N, B \times L, D))$, $v'' = v' + \text{softmax}(\text{matmal}(q, k)) \cdot v'$. The softmax operation models temporal weights and *softly* merges tokens along the temporal dimension. This is distinct from spatial token merging, which employs average pooling and reduces the token count. Here, we use a weighted average pooling along the temporal dimension, maintaining the original token count.

Our approach, depicted in Figure 2, maintains the integrity of pre-existing, well-trained image-text models, thus avoiding the need for model realignment, a requirement in alternative designs that insert a temporal Q-former between the visual encoder and the language model.

## 4 EXPERIMENTS

Our experimental setup is as follows:

- **Pre-training Data** Our model is pre-trained using the MSCOCO (Lin et al., 2014) and CapFilt (Li et al., 2022b) datasets, which include BLIP's pseudo-labeled Conceptual Captioning (Sharma et al., 2018), SBU (Ordonez et al., 2011), and LAION (Schuhmann et al., 2022) datasets—similar to the data sources utilized in BLIP-2. Note that we intentionally exclude the VG (Krishna et al., 2017) dataset from our pre-training procedure, as it mainly consists of localized captions.
- **Models** In order to facilitate a direct and fair comparison with BLIP-2, we employ the same ViT, denoted as `eva-vit-g` (Fang et al., 2022). For the language model decoders, we explore both `opt-2.7b` (Zhang et al., 2022) and `vicuna-7b` (Chiang et al., 2023). Our TomeFormer is initialized using `bert-base-uncased`, ensuring parameter count parity with BLIP-2's Q-former. The only difference in parameterization between our model and BLIP-2 lies in the additional 32 learnable queries present in the latter.
- **Pre-training Details** Our pre-training setup closely mirrors the configurations of BLIP-2. We utilize a maximum learning rate of $1 \times 10^{-4}$ and a minimum learning rate of $1 \times 10^{-5}$. The learning rate follows a schedule that begins with a linear warm-up phase of 5000 steps starting from $1 \times 10^{-6}$ and then transitions to a cosine decay schedule. Weight decay is set at 0.05. The training is conducted with a batch size of 1600, distributed over either $8\times$ A100-80G or $32\times$ V100-32G.
- **Downstream Tasks** SimVLG-Image is evaluated without additional fine-tuning on a variety of tasks, including MSCOCO captioning, VQAv2 (Goyal et al., 2017), GQA (Hudson & Manning, 2019), and OK-VQA (Marino et al., 2019). For video tasks, SimVLG-Video is evaluated on fine-tuned MSR-VTT (Xu et al., 2016) and MSVD (Chen & Dolan, 2011) captioning tasks.

### 4.1 EVALUATION ON IMAGE-TEXT BENCHMARKS

We conducted comparative evaluations between SimVLG and BLIP-2 on multiple image-text benchmarks, including zero-shot VQAv2, GQA, OK-VQA, and MSCOCO captioning. It is essential to note that BLIP-2 demands an extensive Stage-1 pre-training phase involving 250,000 optimization steps. This phase incorporates three distinct loss functions and necessitates multiple forward passes through the model, a process crucial for BLIP-2 to prevent model divergence.

Table 1 summarizes the results of our experiments. Our primary insights can be distilled into the following key points:

- Utilizing the same training set of 104 million image-text pairs and an equal number of optimization steps (250,000), SimVLG consistently outperforms BLIP-2 across nearly all evaluated tasks.
- Remarkably, SimVLG maintains competitive performance even when its training budget is trimmed to approximately one-third of BLIP-2's, specifically 150,000 optimization steps.
- Our experiments show that SimVLG can produce satisfactory results with a significantly reduced training dataset of 11 million image-text pairs, while still undergoing 150,000 optimization steps.
- SimVLG retains its efficacy even when the training budget is restricted to as few as 90,000 steps, demonstrating the model's efficiency and robustness.

**Training Time** In the Stage-1 pre-training phase, BLIP-2 requires considerable time, necessitating multiple forward passes to optimize three separate loss functions. We document the training durations for both BLIP-2 and SimVLG when utilizing eight A100-80G GPUs in Table 2.

Although BLIP-2 significantly reduces training time relative to predecessors like CoCa, it still mandates an extended training duration—approximately ten days. This extensive time commitment limits the feasibility of researchers to investigate various ViT configurations. Most subsequent works based on BLIP-2 continue to use the pre-trained Q-former in conjunction with the `eva-vit-g` model, thereby narrowing the scope of ViT exploration. In contrast, SimVLG significantly trims the training time while maintaining satisfactory performance levels, thus providing researchers with the latitude to explore a wider array of advanced ViTs in future investigations.

Table 1: Comparison of methods on zero-shot VQA and MSCOCO captioning tasks without additional fine-tuning. [†]: We were able to download approximately 81% of LAION-115M and 78% of CCS-14M from the CapFilt dataset. [‡]: BLIP-2 incorporates an additional set of 32 learnable queries, each with a dimension of 768.

| | Models | # pre-train image-text | # trainable params | # training steps | VQAv2 val | GQA test-dev | OK-VQA test | MSCOCO val |
|---|---|---|---|---|---|---|---|---|
| OPT-2.7b | BLIP-2 | 104M[†] | 110M+[‡] | 250k + 80k | 44.6 | 30.6 | 26.0 | 137.7 |
| | SimVLG | 104M | 110M | 250k | **48.4** | **30.9** | **27.2** | **139.1** |
| | SimVLG | 104M | 110M | 150k | 46.9 | 30.8 | 24.8 | 137.0 |
| | SimVLG | 11M | 110M | 150k | 46.3 | 30.0 | 23.0 | 135.1 |
| | SimVLG | 104M | 55M | 90k | 45.9 | 30.6 | 25.8 | 134.0 |
| Vicuna-7b | BLIP-2 | 104M[†] | 110M+[‡] | 250k + 80k | **57.8** | 35.7 | 27.8 | 138.0 |
| | SimVLG | 104M | 110M | 250k | 54.8 | 35.6 | 30.4 | **139.1** |
| | SimVLG | 104M | 110M | 150k | 55.5 | **36.3** | **30.6** | 137.9 |
| | SimVLG | 11M | 110M | 150k | 54.6 | 34.0 | 27.3 | 138.0 |
| | SimVLG | 104M | 55M | 90k | 53.4 | 34.7 | 30.6 | 137.8 |

Table 2: Runtime comparison of BLIP-2 and SimVLG when utilizing OPT-2.7b as the LLM.

| Models | Stage 1 (5k steps) | Stage 2 (5k steps) | # steps | Clock time | MSCOCO |
|---|---|---|---|---|---|
| BLIP-2 | 3 hrs 50 min | 2 hrs 40 min | 330k | 234 hrs | 137.7 |
| SimVLG | - | 2 hrs 45 min | 250k | 133 hrs | 139.1 |
| SimVLG | - | 2 hrs 45 min | 150k | 80 hrs | 137.0 |
| SimVLG (55M) | - | 2 hrs 35 min | 90k | 47 hrs | 136.8 |

## 4.2 EVALUATION OF SIMVLG-VIDEO ON VIDEO CAPTIONING TASKS

We proceed to evaluate the performance of fine-tuned SimVLG-Video models in video captioning tasks, utilizing OPT-2.7b as the language model decoder. Our investigation includes two specific variants of SimVLG-Video: the first is exclusively pre-trained on image data, while the second is further enhanced by pre-training on a corpus of 2 million video-text pairs sourced from the WebVid (Bain et al., 2021) dataset. To provide a comprehensive evaluation, we benchmark SimVLG-Video against five distinct models, described as follows:

- **Baseline (concat)**: This model processes each frame of a video individually and concatenates their visual features to generate a single prompt for the LLM. This method is analogous to the strategy employed in InstructBLIP.
- **Baseline (mean)**: Similar to the concat baseline, this model processes each video frame individually but averages the visual features to create a single prompt for the LLM.
- **Video-LLaMA**: This variant incorporates the BLIP-2 framework and enhances it with an additional temporal Q-former layer. For this evaluation, we focus solely on the vision-language component of Video-LLaMA.
- **VideoChat**: This model extends BLIP-2 by integrating additional Uniformer modules within the ViT architecture and also incorporates learnable temporal queries in its Q-former component.
- **VideoCoCa**: In this model, we adapt the OpenCoCa framework by mlfoundations and augment the existing CoCa architecture with a learnable attentional pooler, resulting in VideoCoCa.

**Evaluation on MSR-VTT** As detailed in Table 3, SimVLG-Video demonstrates superior performance relative to the baseline models, even without the aid of video-text pre-training. This result highlights the effectiveness of our proposed *Temporal Attentive Soft Token Merging* in capturing temporal dynamics. Additionally, we observe an enhancement in performance when incorporating video-text pre-training along with Self-Critical Sequence Training (SCST) (Rennie et al., 2017).

*Temporal Attentive Soft Token Merging* has the distinct advantage of maintaining the integrity of the well-pretrained image-text model (i.e., SimVLG-Image). This contrasts with models such as Video-LLaMA and VideoChat, where the original BLIP-2 architecture is altered, necessitating a complex re-alignment process using video-text pairs. Our empirical analysis indicates that such re-alignment is a non-trivial endeavor. It is worth noting that our VideoCoCa model is at a disadvantage when

Table 3: Comparison of different models' performance on MSR-VTT captioning. Models are pre-trained using 2 million video-text pairs from WebVid dataset, except for image pre-trained SimVLG.

| Models | Image pre-trained | Video pre-trained | CIDEr | BLEU-4 | METEOR | ROUGE |
|---|---|---|---|---|---|---|
| Baseline (concat) | ✓ | ✓ | 65.5 | 44.4 | 31.9 | 64.1 |
| Baseline (mean) | ✓ | ✓ | 67.8 | 47.3 | 32.2 | 65.0 |
| SimVLG-video | ✓ | | 68.4 | 47.6 | 32.4 | 65.3 |
| SimVLG-video | ✓ | ✓ | 69.8 | 48.3 | 32.6 | 65.8 |
| SimVLG-video-scst | ✓ | ✓ | **74.0** | **49.2** | **33.0** | **66.5** |
| Video-LLaMA | ✓ | ✓ | 59.3 | 47.7 | 29.6 | 63.7 |
| VideoChat | ✓ | ✓ | 58.0 | 46.5 | 29.5 | 63.4 |
| VideoCoCa | ✓ | ✓ | 63.0 | 48.5 | 31.4 | 64.8 |

Table 4: Comparison of different models' performance on MSVD captioning.

| Models | CIDEr | BLEU-4 | METEOR | ROUGE |
|---|---|---|---|---|
| Video-LLaMA | 121.2 | 61.6 | 40.3 | 77.8 |
| VideoChat | 118.4 | 64.1 | 41.0 | 78.7 |
| VideoCoCa | 150.9 | 67.7 | 45.3 | 81.9 |
| SimVLG-video | **158.2** | **68.4** | **46.8** | **83.1** |

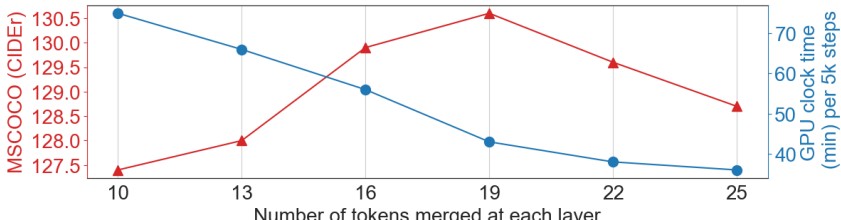

Figure 3: Trade-off between MSCOCO captioning scores (depicted in red) and GPU training time (depicted in blue) as a function of the number of tokens merged ($r$) in TomeFormer. A larger $r$ value leads to shorter soft prompts in the LLM, thereby decreasing computational time (blue line). However, overly compressed soft prompts may result in the loss of valuable visual information, while insufficiently compressed features complicate the optimization process.

benchmarked against Google's reported results, which benefit from extensive training on a much larger dataset of image-text and video-text pairs.

**Evaluation on MSVD**  Similarly, we evaluate SimVLG's performance against Video-LLaMA, VideoChat, and VideoCoCa using the MSVD video captioning dataset, which is presented in Table 4. Our results corroborate that SimVLG consistently surpasses these competing models, further attesting to its robust performance across different video captioning tasks.

## 5 ANALYSIS

**Impact of Soft Prompt Length**  Within the TomeFormer, the vision-to-language connector in SimVLG, we introduce a hyperparameter $r$ that regulates the number of spatial tokens merged at each layer. Increasing $r$ substantially reduces the token count, but runs the risk of eliminating important visual details. On the other hand, a smaller $r$ produces two main effects: (1) a more diffuse representation of visual features, complicating the optimization landscape, and (2) elongated soft prompts for the LLM, leading to increased computational cost during training, such as memory overflow and extended training durations.

To study the effects of $r$, we conduct an ablation experiment using $8\times$ RTX-A6000 and the CCS-14M dataset for pre-training. The models are trained for 60,000 steps, and their performance is evaluated

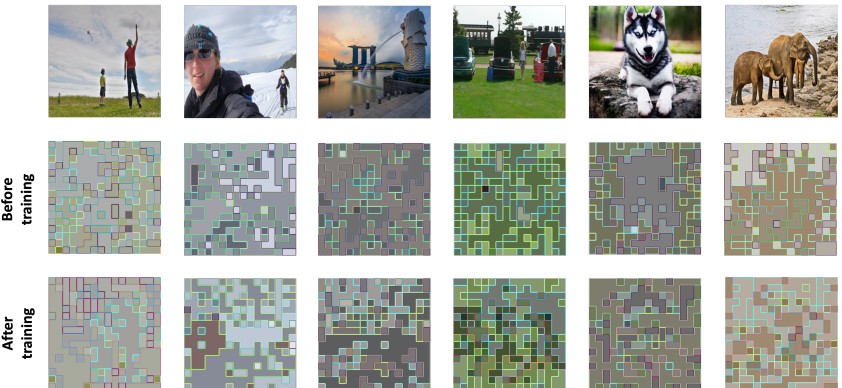

Figure 4: Pre- and post-training visualization of merged tokens in SimVLG. The visual features compressed via token merging exhibit semantic informativeness even prior to training. This inherent characteristic facilitates SimVLG's ability to converge quickly in an end-to-end training setup.

using CIDEr scores on MSCOCO captioning. In Figure 3, we observe that a smaller $r$ (e.g., 10) places a higher computational load on both TomeFormer and the LLM, extending training time and compromising optimization, as evidenced by lower CIDEr scores. In contrast, a larger $r$ value (e.g., 25) expedites training but at the expense of model performance, likely due to excessive feature compression and consequent information loss.

**Token Merging Visualization in SimVLG** One notable advantage of SimVLG over BLIP-2 is the absence of a requisite Stage-1 pre-training for the vision-to-language connector. This simplifies the training pipeline by removing the need to train the model to extract text-informative visual features. We posit that the token merging process in TomeFormer naturally aggregates tokens associated with visually similar elements, thereby yielding concise yet semantically rich visual features from the onset of training. This inherent capability allows SimVLG to benefit from a more streamlined, single-stage training regimen with just one learning objective.

Essentially, our token merging strategy serves as an efficient approximation of QFormer's functionality, compressing visual features in a semantically meaningful manner. Figure 4 illustrates this, displaying the visual tokens before and after training with our TomeFormer. The figure shows that the compressed visual features obtained via token merging are semantically informative and offer basic object segmentation within the image. Furthermore, the semantic coherence of these merged tokens improves as training advances.

## 6 DISCUSSION AND CONCLUSION

This paper introduces SimVLG, an efficient and streamlined pre-training framework for vision-language generative models. Like BLIP-2, SimVLG employs frozen ViT and LLM. It further leverages a conventional Transformer architecture with token-merging capabilities, known as Tome-Former, to act as the vision-to-language connector. Compared to BLIP-2, SimVLG offers the distinct advantage of one-stage training. This reduces computational overhead and maintains competitive performance even with only $1/3$ to $1/6$ of the computational budget required by BLIP-2.

We have also extended SimVLG's applicability to video captioning tasks by incorporating the *Temporal Attentive Soft Token Merging* into its ViT. This enhances the model's temporal modeling capabilities, culminating in the creation of SimVLG-Video. This extension has proven efficacious, delivering commendable performance even without specialized video-text pre-training. Our investigation underscores that a temporal module, which does not disrupt the integrity of the well-pretrained image-text model (e.g., BLIP-2 and SimVLG), is a key factor contributing to this success.

SimVLG demonstrates the possibility of achieving state-of-the-art performance in vision-language tasks without the need for complex training regimens or high computational budgets. This work thus makes a significant contribution to the ongoing efforts to develop more accessible, efficient, and powerful models for understanding and generating visual and textual information.

ETHICS STATEMENT

This research aims to enhance both the efficiency and applicability of vision-language generative models via SimVLG. Although our research does not involve human subjects directly, it is important to acknowledge and discuss the broader ethical implications.

**Data Bias and Fairness:** Our model is trained on publicly available datasets, namely CapFilt, MSCOCO, MSRVTT, MSVD, and WebVid. While these datasets are widely used, we acknowledge that we cannot fully ascertain the extent to which they may contain discriminatory, biased, or sensitive material. Given that our model inherits the biases present in these training datasets, there exists the risk of perpetuating or even amplifying existing societal biases. Despite the broad acceptance of these datasets, caution should be exercised.

**Real-world Deployment and Responsible Usage:** Like all generative models, SimVLG could be misappropriated for creating misleading or harmful content. Thus, it is imperative to implement safety mechanisms to counter such misuse when deploying the model in real-world applications. Special attention should also be paid to ensure that the model does not inadvertently produce outputs that could disclose sensitive or personal information. Finally, while SimVLG is intended as a general-purpose model, its application in contexts that could worsen societal biases or spread misinformation is a pressing concern. Developers and researchers employing SimVLG are advised to be cognizant of these risks and consider incorporating fairness-aware or truth-aware components into their systems.

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
