# OpenReview forum: "SimVLG: Simple and Efficient Pretraining of Visual Language Generative Models"
_ICLR.cc/2024/Conference — ICLR 2024 Conference Withdrawn Submission_

### Official Review · Reviewer_rom5 · 2023-10-30

**Soundness:** 2 fair
**Presentation:** 2 fair
**Contribution:** 2 fair
**Rating:** 3
**Confidence:** 4

**Summary:**

This paper investigates a new framework for pre-training generative vision-language models. It follows the BLIP-2 framework that leverages pretrained frozen vision encoder and LLM. BLIP-2 aligns the vision encoder and LLM by learning a lightweight Q-former in two training stages. This paper argues that the pretraining strategy of BLIP-2 is not efficient and propose a one-stage pretraining method based on token merging. The experiments are conducted on various image and video benchmarks including VQA and captioning.

**Strengths:**

-This paper tackles an important foundamental problem that aims to improves the VLM pretraining efficiency. It proposes to adopt a token merging approach to bridge frozen VIT and LLM. To my knowledge, this idea is new in vision-language model pretraining.

-The experiments are conducted on various downstream tasks including video and image benchmarks.

**Weaknesses:**

### 1. Lacking clarity. I believe the paper is not well written and I could not fully understand the method. In particular, the following technical questions are not clear in the paper.

- 1.1 What is the proposed algorithm to merge tokens exactly? The technical details of TomeFormer do not seem to be presented in the paper.  I was expecting it in Sec. 3.2 but I can only find generic description of the TomeFormer.
- 1.2  The Temporal Attentive Soft Token Merging reshapes the embeddings into shape (N, B×L, D). It means that the self-attention will be calculated over batch and spatial dimension. How does it make sense? Maybe I was missing something.
- 1.3 What is the loss function?

### 2. Experiments are not convincing.

- 2.1 The BLIP-2 numbers in Table 1 are significantly lower than the ones reported in the original BLIP-2 paper (Table 2). The same issue for the numbers of VideoCoCa in Table 3 (see Table 7 in Video CoCa paper). Please properly  explain the difference.
- 2.2 Ablations are missing for token merging in TomeFormer and VIT. Those two are the key technical contribution. It is necessary to show the importance of those two components.
- 2.3 Does it make sense to compare the absolute training time? This paper directly compares the absolute training time with BLIP-2 and claims that their method takes less time. However, the absolute training time does not only depends on the model complexity, but also the optimizer, batch size, hardware, data loading and etc. Why not simply compare FLOPs?
- 2.4 Which metric is used for MSCOCO in Table 1?
- 2.5 Only BLIP-2 is compared in Table 1 on captioning and VQA tasks. Please compare with more recent baselines. The same issue with MSR-VTT benchmark.

**Questions:**

The authors are encouraged to answer my questions above. I would suggest the authors to significantly improve the paper writing and resubmit the paper to another venue. Current form is far from the top conference bar.

---

### Official Review · Reviewer_QWmK · 2023-10-31

**Soundness:** 2 fair
**Presentation:** 2 fair
**Contribution:** 2 fair
**Rating:** 3
**Confidence:** 5

**Summary:**

This paper presents a vision-language pretraining method called SimVLG. The framework is similar to existing methods such as BLIP-2, but extends them such that only one pretraining stage is required. To do that, instead of using the Q-former, SimVLG proposes a tomeformer architecture to compress image tokens as inputs to LLM, so that the total sequence length is relatively short. Experimental results show that the model is competitive to BLIP-2.

**Strengths:**

1. The proposed image token compressor is novel and interesting. This architecture worth future exploration in other applications.

**Weaknesses:**

1. The model is only evaluated on a few tasks, mainly focusing on captioning. It will be more informative to evaluate on other tasks as well, such as image classification, image-text/video-text retrieval.
2. The model is only compared with BLIP-2. I think the key innovation is the tomeformer, so I would suggest to conduct more thorough ablations with alternatives, and study the pro/con of this design against Flamingo.
3. The motivation is not very sound to me: single-stage vs two-stage pretraining used in BLIP-2. It is not mandatory to use 1/2-stage training in either case and it is clear to me why we have to prefer one over the other.

**Questions:**

I would also suggest to consider a different naming for the model. SimVLG is too close to SimVLM, causing a little confusion. Meanwhile, it seems a bit unnecessary to insist to call the model Vision-Language Generative pretraining, as other methods are generative as well (e.g. SimVLM?).

---

### Official Review · Reviewer_KSpy · 2023-10-31

**Soundness:** 3 good
**Presentation:** 3 good
**Contribution:** 2 fair
**Rating:** 5
**Confidence:** 4

**Summary:**

This paper proposes to apply token merging techniques into an adapter architecture to enable efficient training and inference, as well as good performance on downstream tasks, the natural extension to processing video is simple yet effective.

**Strengths:**

1. the token merging-based adapter is efficient and effective.
2. the extension to the video processing is simple yet effective.

**Weaknesses:**

1. incremental idea by combining token merging and Qformer from BLIP.
2. the token merging is not input adaptive, constant r will not be optimal for creating the token with the appropriate length and representability.

**Questions:**

1. In Figure 4, "the visual features compressed via token merging exhibit semantic informativeness even prior to training", however, I did not see the semantic information from these plots in Figure 4, please clarify it in more detail.
2. In Table 1, for Vicuna-7B, the proposed method did not beat the baseline or did not consistently obtain the best performance. Please clarify the reasons.

---

### Official Review · Reviewer_K4ZT · 2023-11-04

**Soundness:** 2 fair
**Presentation:** 3 good
**Contribution:** 1 poor
**Rating:** 3
**Confidence:** 4

**Summary:**

This paper proposes SimVLG, a streamlined framework for the pre-training of computationally intensive V-L generative models. In contrast to the two-stage (representation learning and end-to-end training) scheme in previous V-L generative models, SimVLG only involves one stage and employs a simple language modeling loss for the learning. SimVLG adopts a TomeFormer to alternate the Q-Former, as well as a temporal token merging scheme for video modeling. Experiments show that SimVLG can achieve comparable and even better performances than baselines for both images and videos while reducing the training cost.

**Strengths:**

The high-level goal of this paper (to pursue simple and efficient V-L pre-training) is an interesting and important problem.

**Weaknesses:**

1. Inadequate contributions. From my perspective, the main contributions of this paper are some technical explorations to reduce the training cost of previous V-L models, e.g., BLIP-2. And SimVLG only modifies the Q-Former based on BLIP-2. So I think this is not enough for a conference paper.
2. Incomplete experiments. The proposed one-stage training is quite similar to the second-stage end-to-end training in BLIP-2. So, if you want to demonstrate the efficacy of SimVLG, you should also incorporate the second-stage-only results of BLIP-2 (merging the first-stage data for second-stage training, which has less training cost and is closer to the SimVLG setting). And I conjecture SimVLG may not surpass that.
3. The experimental results are not significant. I agree that SimVLG can speed up the training. However, you claimed that SimVLG can achieve comparable performance with BLIP-2 using only 1/10 data. I cannot agree with this conclusion from Table 1.

**Questions:**

1. The use of *generative* is not justified. *Generative models* typically refer to the models with a low-dim latent space as well as the capacity of decoding upon this space. It is rarely seen that language generation with LLM is referred to as generative models.
2. What are the insights of replacing Q-Former with TomeFormer? I cannot tell the key difference between them. In a nutshell, they are both proposed for the goal of extracting a compact representation from dense input tokens. What exactly makes TomeFormer superior to Q-Former? This is not explained clearly.